# A Wavelet-based Stereo Matching Framework for Solving Frequency Convergence Inconsistency

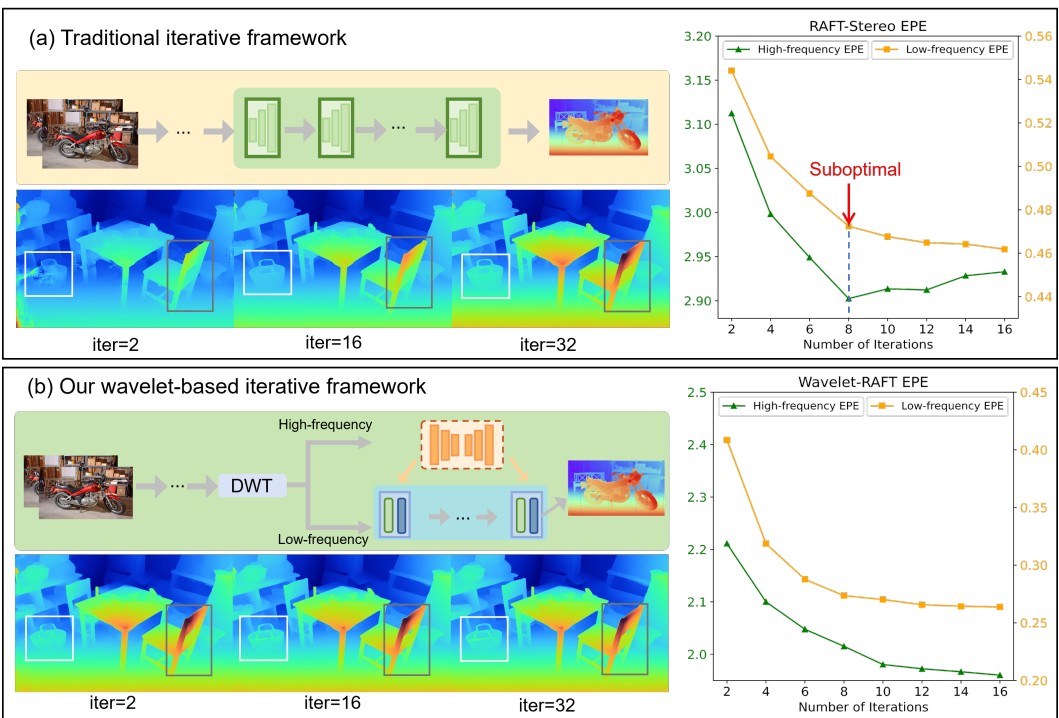

Figure 1: EPE metrics for high and low frequency regions on challenging scenes from the ETH3D dataset (Schops et al., 2017). (a) Traditional iterative-based methods (Lipson et al., 2021) process all frequency components indiscriminately, resulting in inconsistent convergence in different frequency regions. (b) Our frequency-specific modules achieve simultaneous convergence of different frequency components, significantly reducing the required number of iterations. Our method requires only 2 iterations to attain comparable qualitative results to those achieved by the traditional method with 32 iterations.

## Abstract

Through an in-depth analysis the underlying cause of the limited performance in iterative stereo matching methods: **frequency convergence inconsistency**, we propose a novel plug-and-play module named Wavelet-Stereo for this inherent flaw. Specifically, we first summarize the convergence characteristics of distinct frequency components and designed a specialized dual-branch architecture. The high-frequency branch rapidly captures detailed context by a unet, while the low-frequency branch progressively refines the textureless regions throughout the iteration. These two branches interact via a carefully designed high-frequency preservation update operator and predict the disparity, achieving synchronous optimization of both high and low frequency regions. Extensive experiments demonstrate that our Wavelet-Stereo outperforms the state-of-the-art methods and

ranks $1^{st}$ on SceneFlow, ETH3D, KITTI 2015 and KITTI 2012 online leaderboards for almost all metrics. Our work not only uncovers the phenomenon of frequency convergence inconsistency for the first time, but also provides an effective solution and paves the way for new research directions in stereo matching.

# 1 INTRODUCTION

Stereo matching aims to estimate dense disparity maps by matching corresponding pixels between rectified stereo images. This technique serves as the cornerstone for autonomous driving (Yang et al., 2019), augmented reality (Zenati & Zerhouni, 2007), and robotic manipulation (Hsieh & Lin, 2020). Despite decades of research, achieving high-precision and high-efficiency stereo matching remains challenging.

The advent of deep learning has revolutionized the field enabling end-to-end disparity prediction through convolution network (Cheng et al., 2024b; Duggal et al., 2019; Guo et al., 2019; Liang et al., 2019; Nie et al., 2019; Wu et al., 2019; Wei et al., 2025). Aggregation-based methods (Chang & Chen, 2018; Kendall et al., 2017; Shen et al., 2021; Xu & Zhang, 2020) improve accuracy by building 4D correlation volumes and applying 3D convolutions for regularization. To avoid expensive 3D convolution, RAFT-stereo (Lipson et al., 2021) updates the disparity map and hidden states by iteratively indexing from the all-pairs correlation volume and using the gate recursive unit operator. However, the iterative optimization methods (Lipson et al., 2021; Xu et al., 2023) exhibits the following shortcomings: (1) gradual loss of fine-grained information over iteration (Zhao et al., 2023), and (2) struggling to simultaneously capture high-frequency and low-frequency information due to the fixed receptive field (Wang et al., 2024). DLNR (Zhao et al., 2023) designs a decouple module to alleviate the loss of detailed information across the iteration. Selective-Stereo (Wang et al., 2024) employs convolutional kernels with smaller receptive fields to extract high-frequency features, while utilizing larger receptive field kernels for low-frequency features. However, neither of them addresses the essence of these two shortcomings.

Through analysis of RAFT-Stereo's (Lipson et al., 2021) convergence behavior across high-frequency and low-frequency regions, we identify a phenomenon termed **frequency convergence inconsistency** (Figure. 1(a)), i.e., different frequency regions exhibit divergent convergence behaviors during the iteration. We attribute this phenomenon to the expansion of receptive field during the iteration. In early iteration, the model exhibits local receptive fields, allowing it to capture fine-grained details and converge rapidly in high-frequency regions. As iterations progress, the receptive field expands to incorporate broader global context, facilitating convergence in low-frequency regions. However, this enlarged receptive field simultaneously leads to the blurring of fine details, resulting in the degradation in high-frequency regions.

In this paper, we propose a plug-and-play module named Wavelet-Stereo for solving frequency convergence inconsistency. Specifically, we first explicitly decompose the left image into high and low frequency components with the Haar wavelet (Phung et al., 2023). Then, we designed a dual-branch architecture to process high-frequency and low-frequency components separately. Since the high-frequency components exhibit local characteristics, we employ a simple unet network to fully extract the global high-frequency context. For the low-frequency components, we maintain consistency with previous methods and initialize the hidden states with them. Finally, we propose a novel high-frequency preservation update operator (HPU) to prevent the degradation of high-frequency context during the iteration and update the hidden state. The proposed HPU contains two modules: (1) An iterative-based frequency adapter (IFA) can adaptively finetune the global high-frequency context to the iteration-specific high-frequency context based on iteration state. (2) A high-frequency preservation LSTM (HP-LSTM) updates the disparity without propagating the iteration-specific high-frequency context to next iteration, thus preserving detail. As illustrated in Figure. 2, our frequency-specific method excels in challenging scenarios containing fine distant structures. Extensive experiments demonstrate that our Wavelet-Stereo outperforms the state-of-the-art methods and ranks **1$^{st}$ on KITTI 2015 , KITTI 2012, SceneFlow and ETH3D leaderboards** for almost all metrics.

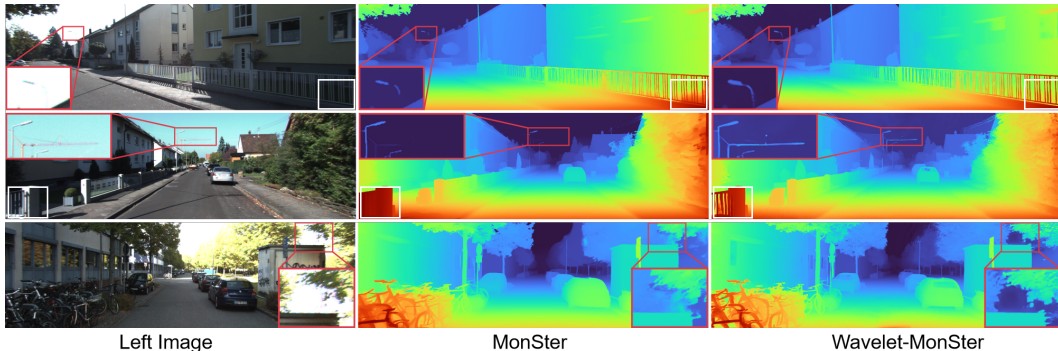

Left Image  MonSter  Wavelet-MonSter

Figure 2: **Visual comparison on KITTI.** All models are trained on Scene Flow and tested directly on KITTI (Geiger et al., 2012; Menze & Geiger, 2015). Our Wavelet-MonSter outperforms MonSter ( (Cheng et al., 2025)) in challenging areas with high-frequency details and fine structures.

## 2 RELATED WORK

**Aggregation-based methods in Stereo Matching.** Aggregation-based methods ( (Chang & Chen, 2018; Cheng et al., 2024a; 2022; Guo et al., 2019; Wei et al., 2025)) have shown significantly improvement in accuracy and robustness. GCNet ( (Kendall et al., 2017)) a 4D correlation volume by concatenating the left and right feature maps within a predefined disparity search range, followed by cost aggregation using 3D convolutions to generate the final matching results. To avoid the use of 3D convolution, AANet ( (Xu & Zhang, 2020)) introduces intra-scale and cross-scale cost aggregation to capture the edge and non-edge area. ACVNet ( (Xu et al., 2022b)) propose the attention concatenation volume to eliminate noise in the cost volume and improve the performance in the ambiguous region.

**Iterative-based methods in Stereo Matching.** Iterative-based methods ( (Chen et al., 2024; Feng et al., 2024; Hu et al., 2021)) have demonstrated significant advantages over aggregation-based methods. RAFT-Stereo ( (Lipson et al., 2021)) introduces an all-pairs correlation volume pyramid and utilizes GRU-based update operators to perform iterative disparity updates. On this basis, IGEV-Stereov ( (Xu et al., 2023)) addresses the issue that the initial correlation volume is excessively coarse by a lightweight cost aggregation network before iteration. CREStereo (Li et al., 2022) proposes a adaptive group correlation layer, computes correlations within local search windows to reduce memory and computational overhead. These methods suffer from slow convergence due to their inability to effectively coordinate the refinement of high and low frequency region.

**Frequency-based methods in Stereo Matching.** Although frequency domain information (Chen et al., 2019; Fritsche et al., 2019) has been widely applied in computer vision tasks, its utilization in the field of stereo matching remains relatively limited. (Yang et al., 2020) learns wavelet coefficients for disparity prediction. Selective-Stereo ( (Wang et al., 2024)) utilizes convolutions with distinct receptive fields to capture high frequency and low frequency context respectively. DLNR ( (Zhao et al., 2023)) proposed a decouple module that separates high-frequency context from hidden states, alleviate the problem of data coupling. However, these method still transfer high-frequency context across the iterations, leading to degradation of high-frequency regions during later stages of the iteration.

## 3 METHODOLOGY

### 3.1 OVERALL PIPELINE

Since our method can be integrated into any iterative-based methods, we use Wavelet-RAFT as a representative example to demonstrate the key innovations of our framework. We employ the same feature extraction network $E_f$ and cost-volume construction as RAFT-Stereo (Lipson et al., 2021) used. As shown in Figure. 3, our framework consists of three steps: (1) **Frequency Decomposition**: we explicitly separate high-frequency and low-frequency components by DWT in Section 3.2. (2) **Frequency Context Extraction**: we extract multi-scale global high-frequency context and low-frequency context separately in Section 3.3. (3) **Iterative Update**: we propose a novel update operator

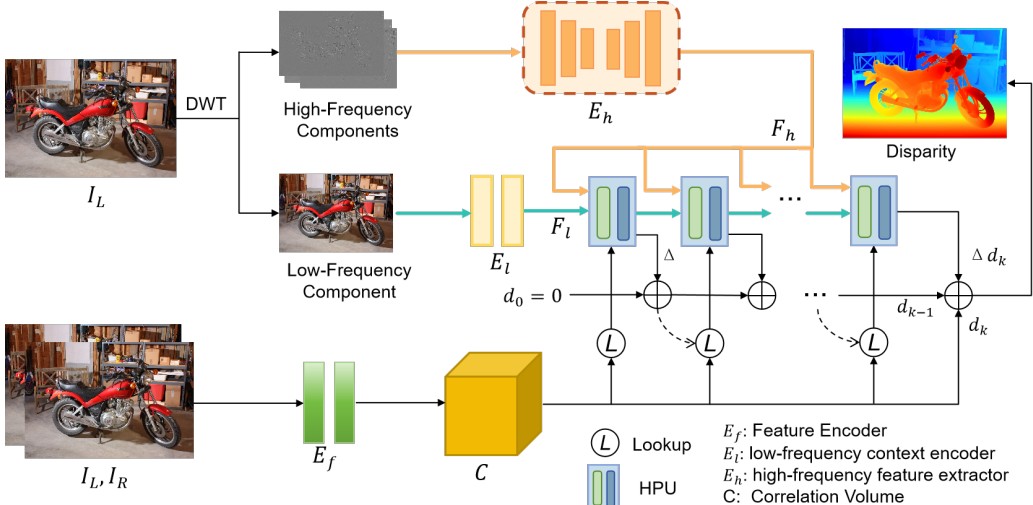

Figure 3: **Overview of Wavelet-RAFT**. Wavelet-RAFT employs a dual-branch architecture comprising: (1) high-frequency branch for capturing global high-frequency context $F_h$, (2) a updating branch that progressively refines hidden states. The global high-frequency context $F_h$ serve as guidance injected into the High-frequency Preservation Update (HPU) operator to update the hidden states during each iteration.

that leverages high-frequency context and low-frequency context to collaborate in each iteration in Section 3.4.

## 3.2 FREQUENCY DECOMPOSITION

We use the Haar wavelet (Phung et al., 2023) to decompose the left image $I_L$ into four sub-images $I_{sub}$ with low and high frequency components, i.e., $I_{sub} = \text{DWT}(I_L)$, where $sub \in \{LL, LH, HL, HH\}$, $I_{LL}$ represents the low-frequency component, and $I_{LH}, I_{HL}, I_{HH}$ correspond to the high-frequency components. To obtain multi-scale frequency components, we repeatedly apply DWT to the low-frequency sub-image ($I_{LL}$), i.e., $I_{sub}^i = DWT(I_{LL}^{i-1})$, where $i \in \{1, ..., n\}$, n is the number of DWT, $I_{sub}^i \in \mathbb{R}^{\frac{H}{2^i} \times \frac{W}{2^i} \times 3}$, and $I_{LL}^0 = I_L$.

## 3.3 MULTI-SCALE FREQUENCY CONTEXT EXTRACTION

We explicitly obtain the high-frequency and low-frequency components of $I_L$ by DWT, which allows us to process them separately according to their respective frequency characteristics.

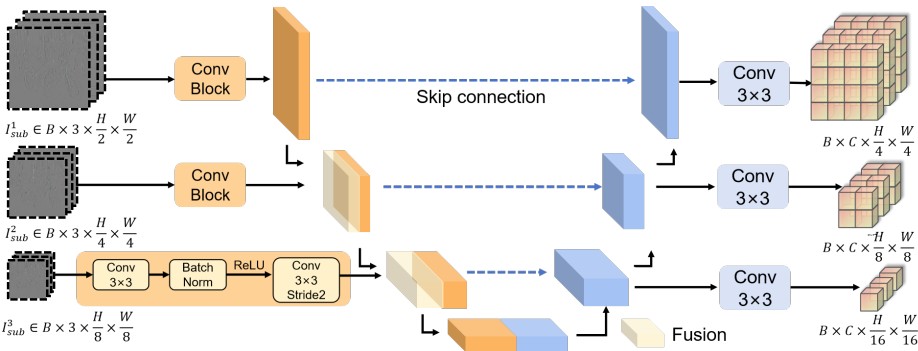

Figure 4: The framework of proposed high-frequency context extractor consisting of a U-shaped network and a series of convolutions blocks, effectively capturing global high-frequency context through multi-scale context aggregation and skip connection.

**Global High-frequency Context Extraction.** To capture global high-frequency details in textures, edges, and thin objects (see the second row of Figure. 2), we design a U-shaped network as the global high-frequency context extractor $E_h$, as shown in Figure. 4. It takes multi-scale high-frequency components $I^i_{sub}$ ($sub \in \{LH, HL, HH\}$) and outputs multi-scale global high-frequency context $F^i_h$ at 1/4, 1/8 and 1/16 resolution, i.e., $F^i_h = E_h(I^i_{sub})$. Due to the localized characteristics of high-frequency components, a lightweight architecture $E_h$ is sufficient to adequately aggregate detailed information.

**Low-frequency Context Extraction.** To capture low-frequency context in smooth regions (see the third row of Figure. 2), we utilize the context encoder in RAFT-Stereo as the low-frequency context extractor $E_l$. The network consists of a series of residual blocks and downsampling layers, producing multi-scale low-frequency context $F^i_l$ at 1/4, 1/8 and 1/16 resolution from low-frequency component $I^1_{LL}$, i.e., $F^i_l = E_l(I^1_{LL})$.

### 3.4 High-frequency Preservation Update Operator

In order to fully fusion the extracted high-frequency and low-frequency context, we propose a novel High-frequency Preservation Update operator (HPU), which consists of Iterative Frequency Adapter (IFA) and High-frequency Preservation LSTM (HP-LSTM), as illustrated in Figure. 5.

$$F^i_l = HPU(F^{global}_h, F^{i-1}_i), i \in [1, 2, 3, ..., k] \tag{1}$$

where k represents the number of HPU iterations.

**Iterative-based Frequency Adapter:** Although the global high-frequency context contains rich detailed information, directly incorporating it into the update operator is suboptimal, as the network requires different information at different iteration stages. To address this, the IFA adaptively fine-tunes the global high-frequency context to iteration-specific high-frequency context based on the current iteration state, i.e., $F'_h = IFA(F^{global}_h)$. Specifically, we design two attention modules to refine frequency-aware features adaptively at each stage (Woo et al., 2018). (1) The Low-frequency Selection Attention (LSA) module generates structural attention maps $A_l$ that incorporate global structure cues into the high-frequency context $F'_h$. (2) The High-frequency Selection Attention (HSA) module produces texture-aware attention maps $A_h$ to enhance the hidden states $F_l$ with fine-grained details.

$$F'^{i,j,k}_h = A^{j-1}_l \odot F'^{i,j-1,k}_h, \quad F^{i,j,k}_l = F^{i,j-1,k}_l, \quad A^{j-1}_l = LSA(F^{i,j-1,k}_l), j \in [1, 3, 5, ...] \tag{2}$$

$$F^{i,j,k}_l = A^{j-1}_h \odot F^{i,j-1,k}_l, \quad F'^{i,j,k}_h = F'^{i,j-1,k}_h, \quad A^{j-1}_h = HSA(F'^{i,j-1,k}_h), j \in [2, 4, 6, ...] \tag{3}$$

where $\odot$ represents elementwise multiplication, $i$ denotes the resolution dimension (1/4, 1/8, and 1/16), $j$ is defined as the iteration number in IFA, while $k$ is defined the number of HPU iterations.

**High-frequency Preservation LSTM:** Obtained the iteration-specific high-frequency context from the IFA, we propose the HP-LSTM to incorporate the finetuned high-frequency context $F'_h$, along with other conditioning such as the correlation volume $C$ and previous disparity $d_{k-1}$), into the update of the current hidden state $F^k_l$. It is worthy that the finetuned high-frequency context $F'^{k-1}_h$ will not be propagated to the next iteration k.

$$F^k_l, \triangle d_k = LSTM_{HP}(F^{k-1}_l | F'^{k-1}_h, L(C, d_{k-1})) \tag{4}$$

where $L$ refers lookup operator, the residual disparity $\triangle d_k$ is decoded from the hidden state $F^k_l$ by a decoder head $D$. The disparity $d$ is updated by

$$d_k = d_{k-1} + \triangle d_k. \tag{5}$$

### 3.5 Loss Function

We use progressively weighted $L_1$ loss across all predicted disparities $\{d_k\}$. Given the ground truth of disparity $d_{gt}$, the total loss is defined as ($\gamma = 0.9$):

$$\mathcal{L} = \sum_{k=1}^{n_k} \gamma^{n_k - i} \|d_k - d_{gt}\|_1. \tag{6}$$

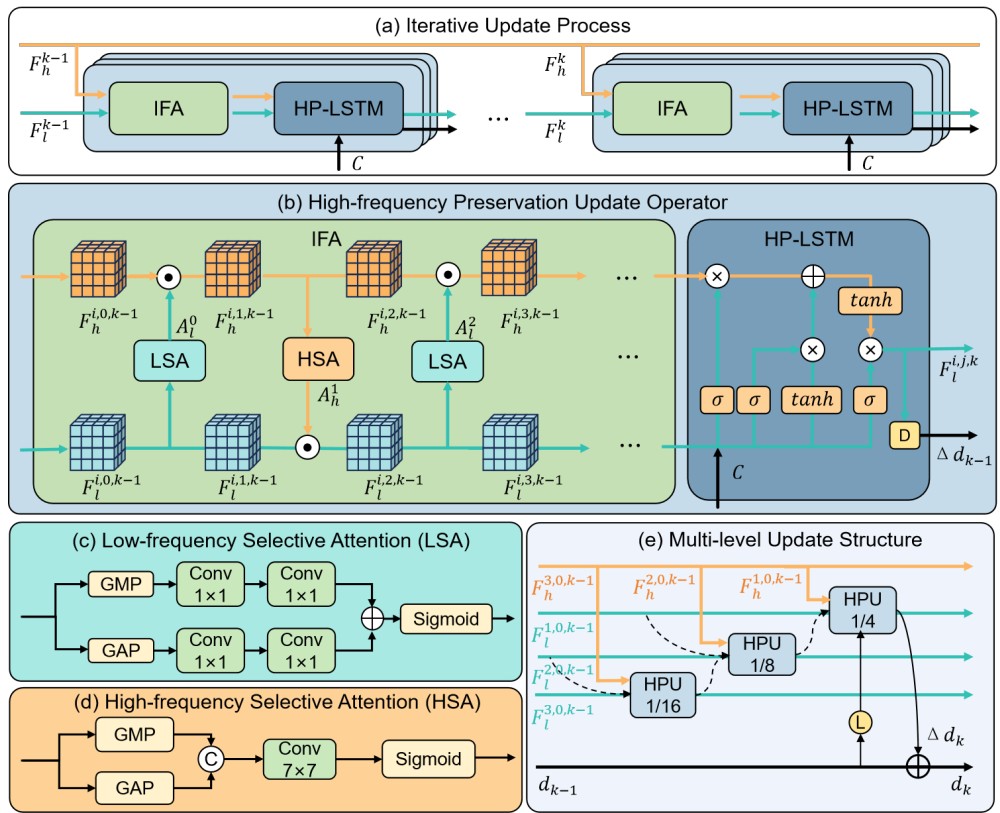

Figure 5: (a) The iterative update process of hidden states $F_l$, guided by global high-frequency $(F_h^{k-1} = F_h^k)$. (b) Proposed high-frequency preservation update operator that finetunes the global high-frequency by iterative-based frequency adapter and update hidden states by high-frequency preservation LSTM. (c) The LSA module adaptively integrates spatial structure information to enhance high-frequency context (d) The HSA module injects details to enrich low-frequency context. (e) Our multi-level update structure to update hidden states from 1/16 to 1/4.

| **Algorithm 1** RAFT-Stereo Pipeline | **Algorithm 2** Our Wavelet-RAFT Pipeline |
|---|---|
| **Require:** a pair of rectified images $I_L, I_R$ | **Require:** a pair of rectified images $I_L, I_R$ |
| 1: $f_L, f_R = E_f(I_L, I_R)$ | 1: $f_L, f_R = E_f(I_L, I_R)$ |
| 2: $C = \text{correlation}(f_L, f_R), d_0 = 0$ | 2: $C = \text{correlation}(f_L, f_R), d_0 = 0$ |
| 3: | 3: $I_{LL}^i, I_{HL}^i, I_{LH}^i, I_{HH}^i = DWT(I_L), i = 1, 2, 3$ |
| 4: $F_l^0 = E_l(I_L)$ | 4: $F_l^0 = E_l(I_{LL}^1)$ |
| 5: | 5: $F_h^{global} = E_h(concat(I_{HL}^i, I_{LH}^i, I_{HH}^i))$ |
| 6: **for** $k = 1, \cdots, n_k$ **do** | 6: **for** $k = 1, \cdots, n_k$ **do** |
| 7:    $F_l^k, \triangle d_k = GRU(F_l^{k-1}, \text{L}(C, d_k))$ | 7:    $F_l^k, \triangle d_k = HPU(F_l^{k-1}, F_h^{global}, \text{L}(C, d_k))$ |
| 8:    $d_k = d_{k-1} + \triangle d_k$ | 8:    $d_k = d_{k-1} + \triangle d_k$ |
| 9: **end for** | 9: **end for** |
| 10: **return** disparity $d$ | 10: **return** disparity $d$ |

## 4 EXPERIMENT

### 4.1 IMPLEMENTATION DETAILS

Wavelet-Stereo is implemented in Pytorch and trained using two NVIDIA A6000 GPUs. For all experiments, we use the AdamW (Loshchilov & Hutter, 2017) optimizer and clip gradients to the range [-1, 1]. We use the one-cycle learning rate schedule with a minimum learning rate of 2e-4. We pretrain Wavelet-Stereo on the Scene Flow dataset (Mayer et al., 2016) with a batch size of 8 and 200k steps. The ablation experiments are trained with a batch size of 6 for 100k steps. We randomly

| Method | RAFT-Stereo | ACVNet | IGEV-Stereo | Wavelet-RAFT (Ours) | MonSter | Wavelet-MonSter (Ours) |
|---|---|---|---|---|---|---|
| EPE (px) | 0.53 | 0.48 | 0.47 | 0.46 | 0.37 | **0.36** |

Table 1: Quantitative evaluation on Scene Flow test set. **Bold**: Best

| | ETH3D | | | KITTI 2015 | | | | KITTI 2012 | | | |
|---|---|---|---|---|---|---|---|---|---|---|---|
| | Bad1.0 Noc | Bad1.0 All | RMSE Noc | D1-fg Noc | D1-all Noc | D1-fg All | D1-all All | Out-2 Noc | Out-2 All | Out-3 Noc | Out-3 All |
| GwcNet (Guo et al., 2019) | 6.42 | 6.95 | 0.69 | 3.49 | 1.92 | 3.93 | 2.11 | 2.16 | 2.71 | 1.32 | 1.70 |
| GANet (Zhang et al., 2019) | 6.22 | 6.86 | 0.75 | 3.37 | 1.73 | 3.82 | 1.93 | 1.89 | 2.50 | 1.19 | 1.60 |
| LEAStereo (Cheng et al., 2020) | - | - | - | 2.65 | 1.51 | 2.91 | 1.65 | 1.90 | 2.39 | 1.13 | 1.45 |
| ACVNet (Xu et al., 2022a) | 2.58 | 2.86 | 0.45 | 2.84 | 1.52 | 3.07 | 1.65 | 1.83 | 2.35 | 1.13 | 1.47 |
| CREStereo (Li et al., 2022) | 0.98 | 1.09 | 0.28 | 2.60 | 1.54 | 2.86 | 1.69 | 1.72 | 2.18 | 1.14 | 1.46 |
| IGEV-Stereo (Xu et al., 2023) | 1.12 | 1.51 | 0.34 | 2.62 | 1.49 | 2.67 | 1.59 | 1.71 | 2.17 | 1.12 | 1.44 |
| CroCo-Stereo (Weinzaepfel et al., 2023) | 0.99 | 1.14 | 0.30 | 2.56 | 1.51 | 2.65 | 1.59 | - | - | - | - |
| Selective-IGEV (Wang et al., 2024) | 1.23 | 1.56 | 0.29 | 2.55 | 1.44 | 2.61 | 1.55 | 1.59 | 2.05 | 1.07 | 1.38 |
| LoS (Li et al., 2024) | 0.91 | 1.03 | 0.31 | 2.66 | 1.52 | 2.81 | 1.65 | 1.69 | 2.12 | 1.10 | 1.38 |
| NMRF-Stereo (Guan et al., 2024) | - | - | - | 2.90 | 1.46 | 3.07 | 1.57 | 1.59 | 2.07 | 1.01 | 1.35 |
| DEFOM-Stereo (Jiang et al., 2025) | 0.70 | 0.78 | 0.22 | 2.24 | 1.33 | 2.23 | 1.41 | 1.43 | 1.79 | 0.94 | 1.18 |
| MonSter (Cheng et al., 2025) | 0.46 | 0.72 | 0.20 | 2.76 | 1.33 | 2.81 | 1.41 | 1.36 | 1.75 | 0.84 | 1.09 |
| Wavelet-MonSter(ours) | 0.35 | 0.63 | 0.18 | 2.60 | 1.31 | 2.60 | 1.38 | 1.32 | 1.71 | 0.83 | 1.07 |

Table 2: Results on three popular benchmarks. All results are derived from official leaderboard publications or corresponding papers. All metrics are presented in percentages, except for RMSE, which is reported in pixels. For testing masks, "All" denotes testing with all pixels while "Noc" denotes testing with a non-occlusion mask. The best and second best are marked with colors.

crop images to $320 \times 736$ and use the same data augmentation as (Lipson et al., 2021) for training. We use 22 update iterations during training and 32 updates for evaluation. The pipeline comparison of traditional iterative-based framework with ours is shown in Algorithm 1 and Algorithm 2.

## 4.2 BENCHMARK DATASETS AND PERFORMANCE

We evaluate Wavelet-Stereo on four widely used benchmarks and submit the results to online leaderboards for public comparison: KITTI 2012 (Geiger et al., 2012), KITTI 2015 (Menze & Geiger, 2015), ETH3D (Schops et al., 2017), and Scene Flow (Mayer et al., 2016).

**Scene Flow**. To verify the universality of our proposed framework, we take RAFT-Stereo and MonSter as baseline and integrate our framework. As shown in Table. 1, both of our models surpass its baseline and our Wavelet-MonSter establishing a new state-of-the-art EPE benchmark on Scene Flow. To validate the ability to handle different frequency regions, we split Scene Flow test set into high-frequency and low-frequency region. As shown in Table. 4, quantitative comparisons reveal that our Wavelet-Raft outperforms Selective-RAFT (Wang et al., 2024) on EPE metric and surpasses the baseline by 22%. Compared to Selective-IGEV and DLNR (Zhao et al., 2023) which is designed for frequency issues, our Wavelet-MonSter outperforms them by 25.89% and 10.3% in high-frequency regions, 30.2% and 13.7% in low-frequency regions, respectively.

**ETH3D**. Following MonSter (Cheng et al., 2025), we firstly finetune the Scene Flow pretrained model on the mixed Tartan Air (Wang et al., 2020), CREStereo Dataset (Li et al., 2022), Scene Flow (Mayer et al., 2016), Sintel Stereo (Butler et al., 2012), InStereo2k (Bao et al., 2020) and ETH3D (Schops et al., 2017) datasets for 300k steps. Then we finetune it on the mixed CREStereo Dataset, InStereo2k and ETH3D datasets with for another 90k steps. As shown in Table. 2, our Wavelet-MonSter outperforms MonSter by 24.9% on Bad 1.0 metric, and rank $1^{st}$ among all methods under identical configurations.

**KITTI**. Following the training of MonSter (Cheng et al., 2025), we finetune our pretrained model on the mixed dataset of KITTI 2012 (Geiger et al., 2012) and KITTI 2015 (Menze & Geiger, 2015) with a batch size of 8 for 50k steps. For best performance, we evaluate our Wavelet-MonSter on the test set of KITTI 2012 and KITTI 2015, with results submitted to the official KITTI online leaderboard. As shown in Table. 2, our Wavelet-MonSter achieves the best performance among all published approaches to date and ranks $1^{st}$ on both the KITTI 2015 and KITTI 2012 leaderboards for almost all metrics, outperforming over 280 competing methods. Figure. 6 shows qualitative results on KITTI 2012 and KITTI 2015 test sets, where our Wavelet-MonSter significantly outperforms MonSter in both detailed high-frequency regions (see the first and second row of figure) and **reflective regions with complex textures** (see the third row of figure) in the difficult scenarios.

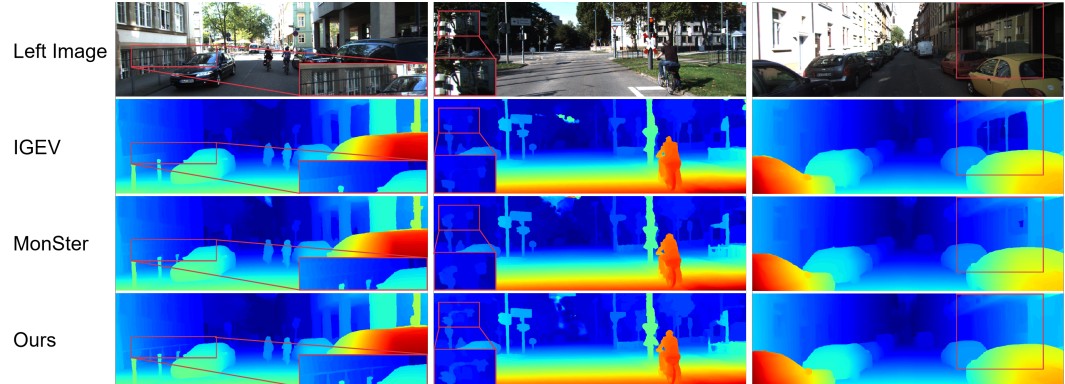

Figure 6: Qualitative results on KITTI test set. Our Wavelet-MonSter outperforms MonSter in challenging areas with high-frequency details and weak texture.

### 4.3 ABLATION STUDY

We conducted comprehensive ablation studies to validate the contribution of each component in our framework. Due to the simplified training settings, the quantitative results of ablation experiments differ from the comparison results described above. We present the main results of ablation experiments, and more results can be found in Appendix A.

**Effectiveness of proposed modules.** The results in the Table. 3 demonstrate that it is effective and necessary to propose frequency-specific module for features with distinct convergence characteristics. To assess the importance of the high-frequency context extractor $E_h$ , we replace $E_h$ with a simple two-layer convolutional network. Quantitative results (EPE increases from 0.52 to 0.56) demonstrate that a powerful context extraction network is needed to adequately fuse high-frequency information at multiple scales.

To validate the effectiveness of HPU and its components, we conducted ablation studies by removing or replacing key elements of the module.

First, we remove the IFA component from the HPU, which means directly incorporating the global high-frequency context $F_h^{global}$ into the updating process without finetuning. This modification lead to a significant degradation in performance (EPE increases from 0.52 to 0.56) , underscoring the necessity of adaptively refining high-frequency context before it is used to update hidden states.

Second, we replace the HP-LSTM with a standard LSTM, which means the fine-tuned high-frequency context $F_h'$ is transfered into the next iteration without preservation. The degradation in performance (EPE increases from 0.52 to 0.53) precisely validated our motivation to preserve high-frequency context during the iterations.

| Model | $E_h$ | IFA | HP-LSTM | GRU | EPE (px) | D1 (%) |
|---|---|---|---|---|---|---|
| Baseline (RAFT-Stereo) | | | | ✓ | 0.62 | 8.40 |
| w/o HP-LSTM | ✓ | ✓ | | | 0.53 | 6.34 |
| w/o IFA | ✓ | | ✓ | | 0.56 | 6.64 |
| w/o HPU | ✓ | | | ✓ | 0.58 | 7.29 |
| w/o $E_h$ | | ✓ | ✓ | | 0.56 | 6.72 |
| Full model (Wavelet-RAFT) | ✓ | ✓ | ✓ | | **0.52** | **6.21** |

Table 3: Ablation study of the effectiveness of proposed modules on Scene Flow test set. HPU denotes High-frequency Preservation Update operator, $E_h$ denotes global high-frequency context extractor, IFA denotes Iterative-based Frequency Adapter, HP-LSTM refers High-frequency Preservation LSTM and GPU refers updating units in RAFT-Stereo

| Method | HFR | LFR |
|---|---|---|
| RAFT-Stereo | 34.00 | 0.72 |
| Selective-RAFT | 27.89 | 0.57 |
| **Wavelet-RAFT** | **26.48** | **0.56** |
| DLNR | 31.60 | 0.63 |
| Selective-IGEV | 26.10 | 0.51 |
| MonSter | 26.08 | 0.47 |
| **Wavelet-MonSter** | **23.42** | **0.44** |

Table 4: Quantitative evaluation on Scene Flow test set in different regions (EPE). HFR refers to the high-frequency region, while LFR refers to the low-frequency region.

| Model | Iteration | EPE | Runtime (s) |
|---|---|---|---|
| RAFT-Stereo | 32 | 0.53 | 0.38 |
| Wavelet-RAFT | 32 | **0.46** | 0.68 |
| | 16 | 0.47 | 0.45 |
| | 8 | 0.52 | **0.23** |

Table 5: Ablation study of the number of iterations.

Additionally, We substituted the HPU with a standard GRU module in RAFT-Stereo. This modification results in performance degradation across all metrics (EPE increases from 0.52 to 0.58 and D1 increases from 6.21 to 7.29). Overall, these experiments substantiate the design motivations of the HPU and highlight the critical roles played by both the IFA and the HP-LSTM in coordinating the convergence of high and low frequency regions.

**Number of Iterations.** As evidenced by Table. 5, our framework significantly accelerates convergence. This improvement stems from mitigating the inherent conflict between high and low frequency components during the iteration, which enables superior performance with markedly fewer iterations than traditional methods. Specifically, our Wavelet-RAFT requires only 8 iterations to surpass the performance of RAFT-Stereo while reducing runtime by 39.5%.

## 5    ZERO-SHOT GENERALIZATION

Robust zero-shot generalization ability is critical for practical stereo matching model. We validate the effectiveness of our Wavelet-Monster by training it solely on the Scene Flow dataset and then testing it on the real-world datasets KITTI 2012, KITTI2015, Middlebury 2014 and ETH3D training sets. As evidenced by the quantitative results in Table. 6, our approach outperforms all comparable methods. Quantitative results on KITTI 2012 and KITTI 2015 training sets in Figure. 2 further substantiates these findings, showing enhanced performance in challenging areas such as textureless surfaces and detailed object boundaries.

| Method | KITTI-12 | KITTI-15 | Middlebury | ETH3D |
|---|---|---|---|---|
| RAFT-Stereo  (Lipson et al., 2021) | 5.12 | 5.74 | 9.36 | 3.28 |
| CREStereo  (Li et al., 2022) | 5.03 | 5.79 | 12.88 | 8.98 |
| Selective-IGEV  (Wang et al., 2024) | 5.64 | 6.05 | 12.04 | 5.40 |
| NMRF-Stereo  (Guan et al., 2024) | 4.23 | 5.10 | 7.54 | 3.82 |
| IGEV-Stereo  (Xu et al., 2023) | 4.84 | 5.51 | 6.23 | 3.62 |
| MonSter  (Cheng et al., 2025) | 3.62 | 3.97 | 5.17 | 2.03 |
| Wavelet-MonSter | **3.37** | **3.56** | **4.74** | **1.86** |

Table 6: **Zero-shot generalization benchmark**. All models are trained on Scene Flow. The 3-pixel error rate is used for KITTI, 2-pixel error rate for Middlebury 2014, and 1-pixel error rate for ETH3D.

## 6    CONCLUSION

Our analysis reveals that the performance degradation in high-frequency regions is a direct consequence of receptive field expansion over iterations. To address this, we introduce Wavelet-Stereo, a plug-and-play module that applies dedicated process to different frequency components. This method fully leverages the convergence properties of different frequency components, avoiding the inherent convergence conflicts in the iterative process of current iterative paradigm methods.

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

# A APPENDIX

## A DATASET AND EVALUATION METRICS

**Pretrain dataset:** Scene Flow (Mayer et al., 2016) is a synthetic stereo matching dataset consisting of 35,454 training image pairs and 4,370 testing image pairs, with a resolution of 960×540. It provides dense disparity maps as ground truth annotations for each image pair. All models in this work are trained exclusively on the SceneFlow training dataset.

**Zero-shot and finetune datasets:** To validate the generalization capability of our model, we evaluate its performance on the training sets of the following four real-world datasets. **KITTI 2012** (Geiger et al., 2012) and **KITTI 2015** (Menze & Geiger, 2015) are real-world driving scene datasets. Specifically, KITTI 2012 provides 194 training pairs and 195 testing pairs, while KITTI 2015 offers 200 training pairs and 200 testing pairs. **ETH3D** (Schops et al., 2017) consists of gray-scale stereo pairs acquired from diverse indoor and outdoor scenes, comprising 27 pairs for training and 20 pairs for testing. **Middlebury** (Scharstein et al., 2014) provides 15 training pairs and 15 testing pairs of high-resolution stereo images captured in indoor environments.

**Metrics:** As usual, we use end-point-error (EPE) and kpx for Scene Flow datasets evaluation metrics, where EPE is the average $l_1$ distance between the prediction and ground truth disparity. And kpx denotes the percentage of outliers with an absolute error greater than k pixels. Referencing previous studies, the thresholds set for each dataset are as follows: 3 pixels for KITTI-2012 and KITTI-2015, 2 pixels for Middlebury, and 1 pixel for ETH3D.

## B IMPLEMENTATION

### B.1 IMPLEMENTATION DETAILS

Following (Lipson et al., 2021), all models are trained with the Adam optimizer ($\beta_1 = 0.9, \beta_2 = 0.999$). For data augmentation setting, the image saturation was adjusted between 0 and 1.4, the right image was perturbed to simulate imperfect rectification that is common in datasets such as ETH3D and Middlebury. We froze all the batch normalization layers in training process. The maximum disparity $D$ for training and evaluation is set to $D = 192$.

### B.2 FREQUENCY CONVERGENCE INCONSISTENCY EXPERIMENT

To quantitatively evaluate frequency-specific performance, we generate edge masks using the Canny operator (implemented via OpenCV, lower=100, upper=200) on ground truth (GT) images for explicit separation of high-frequency regions and low-frequency regions. The binary edge map M serves as a high-frequency region mask, enabling calculation of high-frequency endpoint error (EPE) through element-wise multiplication:

$$EPE_{high} = M \odot EPE_{total} \tag{7}$$

Conversely, $(1 - M)$ serves as a low-frequency region mask and the low-frequency error is computed using the inverted mask $(1 - M)$:

$$EPE_{low} = (1 - M) \odot EPE_{total} \tag{8}$$

### B.3 STRUCTURE OF HIGH-FREQUENCY PRESERVATION UPDATE OPERATOR

The High-frequency Preservation Update Operator is consisted of Iterative-based Frequency Adapter and High-frequency Preservation LSTM.

For Iterative-based Frequency Adapter, it contains two frequency attention module: A low-frequency selection attention (LSA) module and a high-frequency selection attention (HSA) module. The LSA module processes low-frequency features carrying global structural information through a dual-path architecture. Let $F_l \in R^{H \times W \times C}$ denote the input low-frequency feature map. The module first applies both Global Max Pooling (GMP) and Global Average Pooling (GAP) along spatial dimensions to obtain channel-wise features. These pooled features then undergo channel transformation via 1×1

convolutions ($W_1, W_2 \in R^{C \times C}$) followed by $ReLU$ activation function:

$$z_{max} = ReLU[W_1(GMP(F_l))]$$
$$z_{avg} = ReLU[W_2(GAP(F_l))] \qquad (9)$$
$$A_L = \sigma(z_{max} + z_{avg})$$

where $\sigma$ denotes the sigmoid activation function.

The HSA module targets high-frequency patterns containing local textures and details. It employs identical pooling operations but processes them through a 7×7 convolutional layer $W_3$ to capture broader spatial contexts while suppressing noise:

$$A_H = \sigma(W_3(Concat(z_{max}, z_{avg}))) \qquad (10)$$

where $\sigma$ denotes the sigmoid activation function and Concat denotes concatenating along the channel dimension. The LSA module provides global structural context to guide high-frequency processing, while the HSA module supplies local texture details to enrich low-frequency representations.

For the High-frequency Preservation LSTM, it takes high-frequency feature $F_h$ as condition priors along with cost volume C, disparity $d_k$ to update the hidden states $F_l$:

$$x_k = [\text{Encoder}_g(\mathbf{C}), \text{Encoder}_d(d_k), d_k]$$
$$i_t = \sigma(\text{Conv}([h_{k-1}, x_k], W_i) + b_{hi})$$
$$f_t = \sigma(\text{Conv}([h_{k-1}, x_k], W_f) + b_{hf})$$
$$g_t = \tanh(\text{Conv}([h_{k-1}, x_k], W_g) + b_{hg}) \qquad (11)$$
$$o_t = \sigma(\text{Conv}([h_{k-1}, x_k], W_o) + b_{ho})$$
$$c_t = f_t * F_h + i_t * g_t$$
$$F_l = o_t * \tanh(c_t)$$

## C  MORE QUANTITATIVE RESULTS

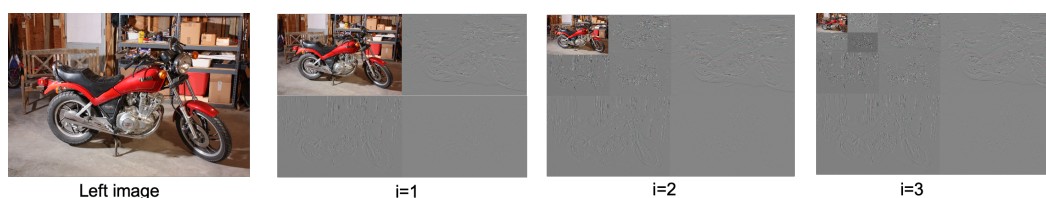

Figure 7: Different level of DWT decomposition (i refers DWT level).

**Effectiveness of multi-scale high-frequency feature extractor** To evaluate the efficacy of our multi-scale high-frequency feature extractor $F_h$, we conduct comprehensive ablation studies by feeding multi-level DWT outputs (Figure. 7) into the module. It introduces only a minimal parameter increase through an efficient fusion module that aggregates multi-level high-frequency features from DWT decomposition. Quantitative evaluation on the Scene Flow test set (Table. 7) demonstrates that this lightweight design adds just 0.77M additional parameters while achieving 2.3% improvements (EPE decreases from 0.483 to 0.472).

Table 7: Ablation studies of the effectiveness of our multi-scale high-frequency feature extractor. 1, 2, 3 stand for the level of Discrete Wavelet Transform (DWT).

| Method | EPE | D1 | Params. (M) |
|---|---|---|---|
| HPU | 0.563 | 6.92 | 0.55 |
| HPU + HAM$_1$ | 0.483 | 6.39 | 4.36 |
| HPU + HAM$_2$ | 0.472 | 6.26 | 4.73 |
| HPU + HAM$_3$ (Ours) | **0.467** | **6.21** | **5.5** |

Our high-frequency feature extractor which is fed 3-level DWT decomposition outputs achieves effective fusion and utilization of multi-scale high-frequency features. This carefully balanced design maintains the model's compactness and practical deployability while enabling effective multi-scale high-frequency feature utilization.

**Number of IFA iteration.** To determine the most appropriate interaction iteration in IFA, we conduct a systematic investigation of IFA interaction rounds by varying j from 1 to 6. As quantified in Table.8, performance exhibits a clear peak at r=4 iterations, with both under-interaction (j<4) and over-interaction (j>4) leading to degraded results. This suggests: (1) sufficient rounds are necessary for finetuning the iteration-specific high-frequency features, yet (2) excessive iterations may cause feature over-smoothing.

| Rounds (j) | EPE | Runtime(s) |
|:---:|:---:|:---:|
| 1 | 0.394 | 0.680 |
| 2 | 0.383 | 0.686 |
| 3 | 0.371 | 0.772 |
| 4 | **0.367** | 0.790 |
| 5 | 0.371 | 0.865s |
| 6 | 0.373 | 0.875s |

Table 8: Ablation study of the rounds(j) in IFA.

**Parameter and Computational Analysis** We further provide quantitative results on memory usage and computational cost. We use a single Nvidia A6000 graphics card (with 48 GiB memory) and the batch size is set to 1 for the inference (16 iterations). The memory consumption and computational cost is shown in Table.9

Table 9: Computational complexity breakdown per stage. Runtime, GPU memory usage, number of parameters, and equivalent FPS are reported.

| Stage | Memory(MB) | Params(M) | Runtime(ms) |
|:---|:---:|:---:|:---:|
| DWT | 0 | - | 33.31 |
| Low-frequency Feature Extraction | 1660 | 4.32 | 10.65 |
| High-frequency Feature Extraction | 2064 | 7.01 | 5.44 |
| Cost volume | 2072 | - | 70.54 |
| HPU-Refinement | 2178 | 6.47 | 369.16 |

## D   MORE QUALITATIVE RESULTS

In this section, we provide a comprehensive qualitative comparison between our method and the baselines on four widely used real-world datasets (KITTI 2012 Geiger et al. (2012), KITTI 2015 Menze & Geiger (2015), Middlebury Scharstein et al. (2014) and ETH3D Schops et al. (2017)). As shown in Figure. 10, Figure. 8, Figure. 11 and Figure. 9, our Wavelet-RAFT exhibits significantly superior zero-shot generalization performance compared to baseline model when pretrained exclusively on the synthetic SceneFlow  (Mayer et al., 2016) dataset. Our Wavelet-MonSter demonstrates remarkable performance in preserving hierarchical details in the predicted disparity maps, with even the most delicate structures being accurately maintained, as shown in Figure. 12.

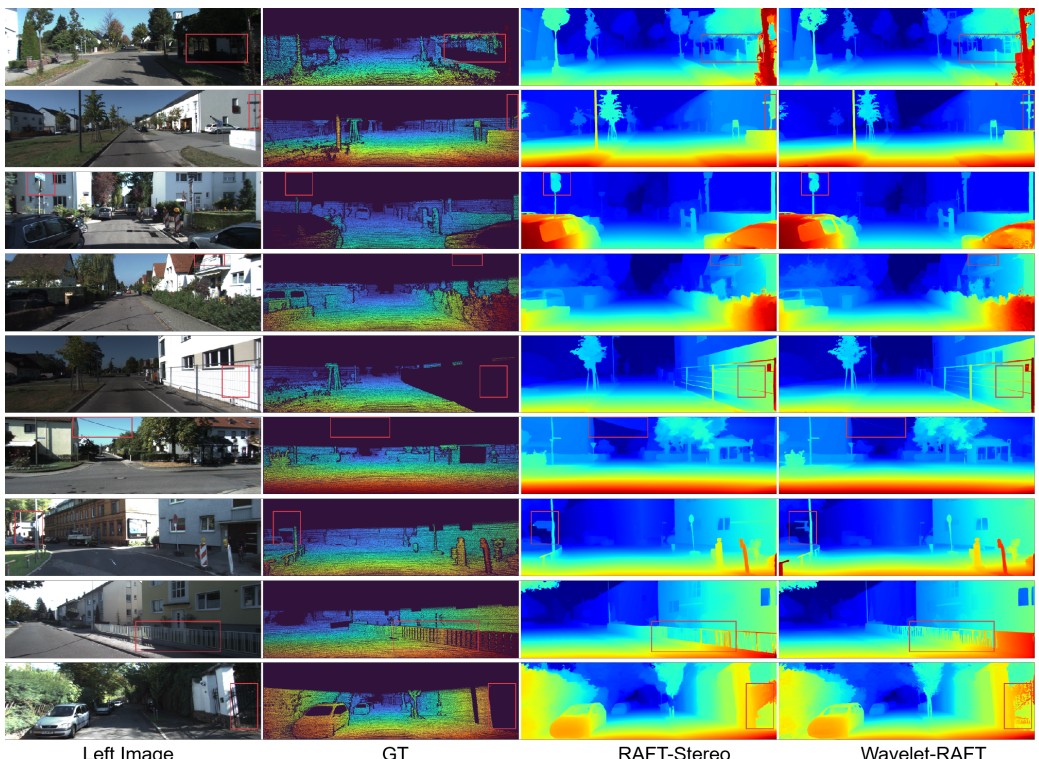

Left Image          GT          RAFT-Stereo          Wavelet-RAFT

Figure 8: **Qualitative Results – Zero-Shot Generalization on the KITTI 2012 and KITTI 2015 train sets**.

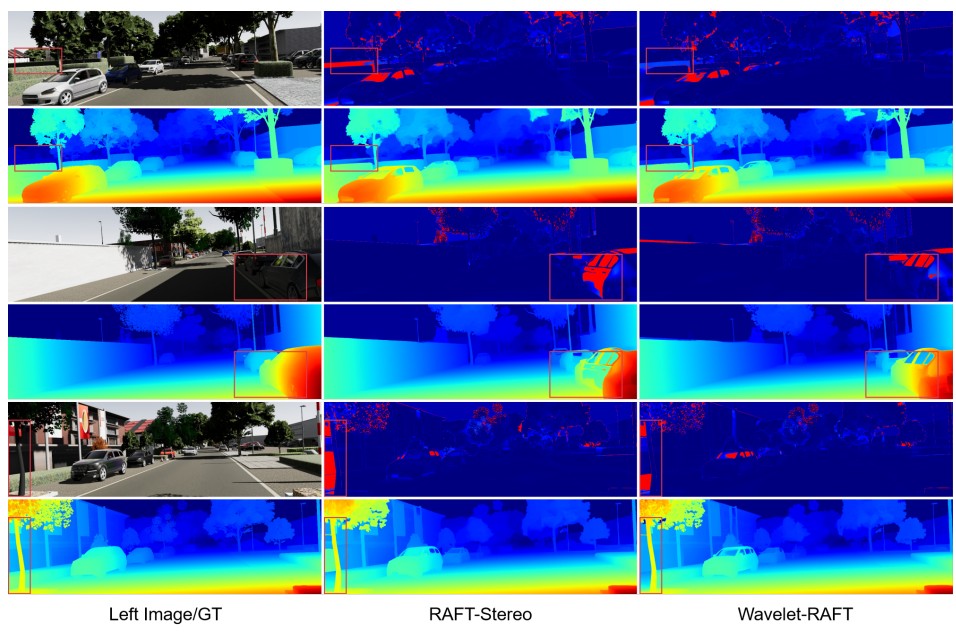

Left Image/GT          RAFT-Stereo          Wavelet-RAFT

Figure 10: Qualitative results on VKITTi train set. The first column shows the left image and the corresponding ground-truth disparity map. The rest columns show the error map and the predicted disparity map of RAFT-Stereo and Wavelet-RAFT, respectively.

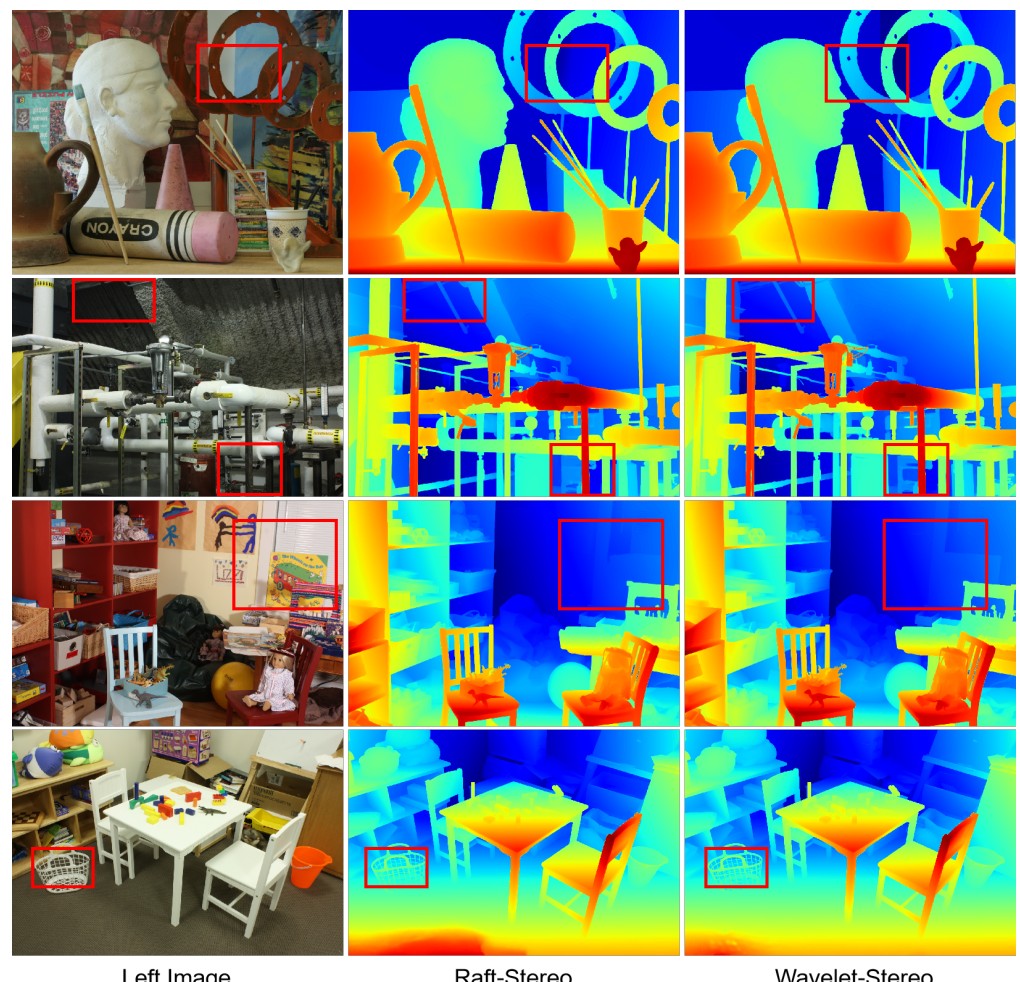

| Left Image | Raft-Stereo | Wavelet-Stereo |

Figure 9: Qualitative Results – Zero-Shot Generalization on the Middlebury Scharstein et al. (2014) train set.

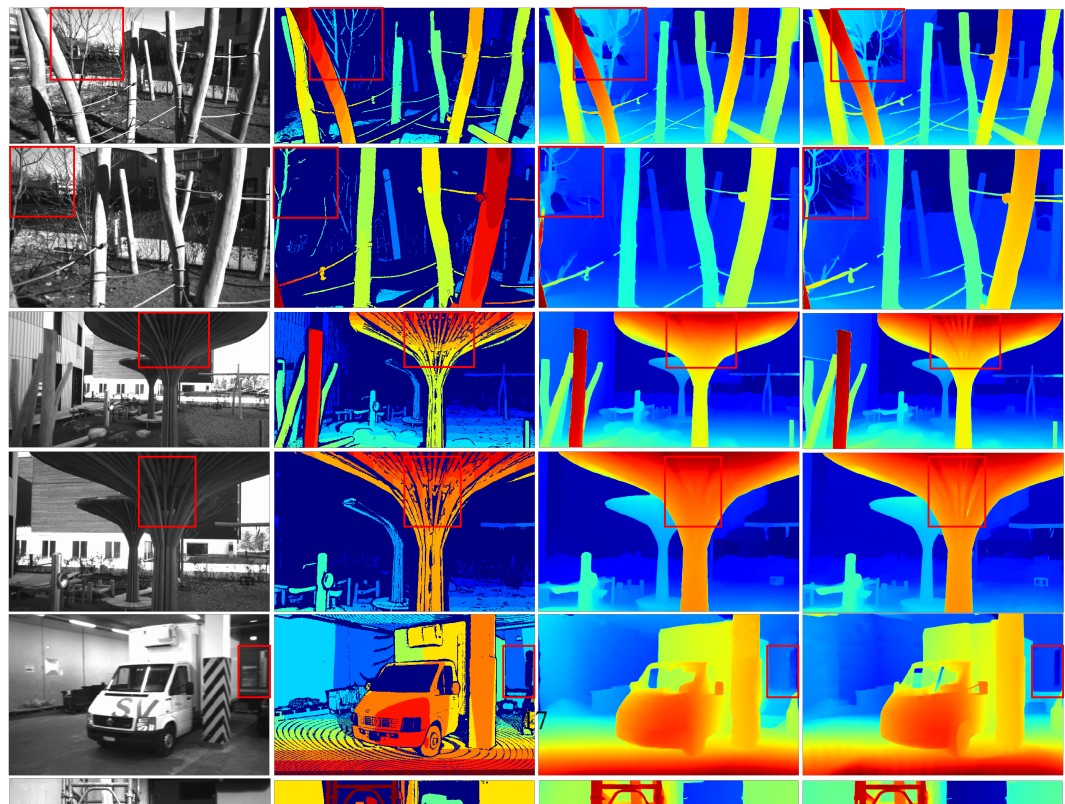

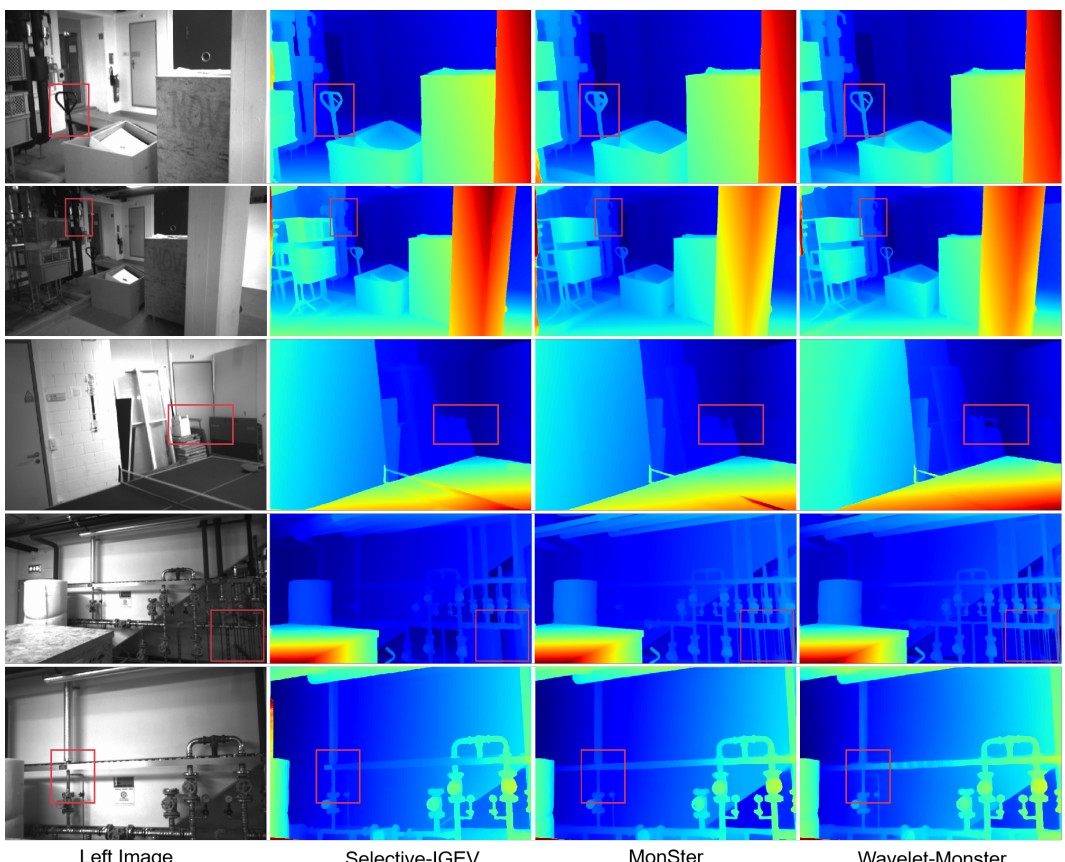

Left Image       Selective-IGEV       MonSter       Wavelet-Monster

Figure 12: Qualitative results on ETH3D test set.

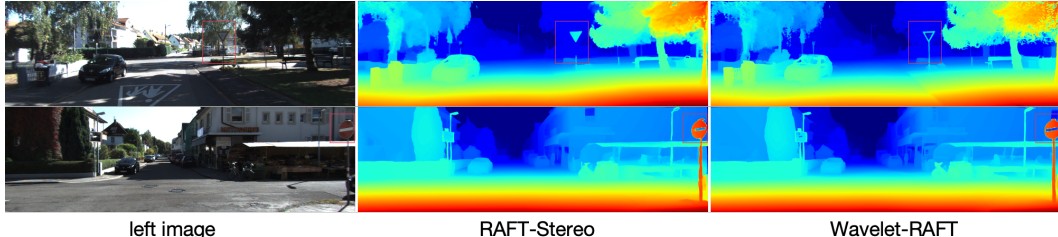

| left image | RAFT-Stereo | Wavelet-RAFT |

Figure 13: Examples of failure cases for the proposed method. Poor performance due to unnecessary extraction of task-irrelevant information.

## E  DISCUSSIONS, LIMITATIONS, AND FURTHER WORK

**Limitations.** While the proposed method demonstrates strong performance, the computational overhead induced by the DWT decomposition, multi-scale feature extraction, and iterative frequency adapter (IFA) operations could potentially hinder real-time deployment. These limitations highlight important trade-offs between frequency-aware precision and computational practicality that warrant further investigation in future work.

**Further Work.** Here are some directions of our future work.

1. Semantics-guided high-frequency processing pipeline that discriminatively extracts task-relevant high-frequency information.

2. Adaptive number of iteration for different scenarios.

3. Exploring the application of diffusion model in stereo matching.

## F  THE USE OF LARGE LANGUAGE MODELS

The authors confirm their full accountability for the scholarly validity and originality of this manuscript. We attest that artificial intelligence was in no way used to generate or falsify research data. The only application of Large Language Models was to aid in wording and phrasing, with the goal of improving the prose's idiomatic flow and making the presentation more accessible to an international academic audience. The final responsibility for the intellectual content and its expression remains entirely with the authors.

