# OpenReview forum: "A Wavelet-based Stereo Matching Framework for Solving Frequency Convergence Inconsistency"
_ICLR.cc/2026/Conference — ICLR 2026 Conference Withdrawn Submission_

### Official Review · Reviewer_2bQC · 2025-10-19

**Soundness:** 2
**Presentation:** 2
**Contribution:** 2
**Rating:** 4
**Confidence:** 4

**Summary:**

This paper identifies and addresses a fundamental issue in iterative stereo matching methods called frequency convergence inconsistency, where high-frequency and low-frequency regions converge at different rates during optimization. The authors propose Wavelet-Stereo, a plug-and-play module that uses wavelet decomposition to separate and process high- and low-frequency components via a dual-branch architecture. A key innovation is the High-frequency Preservation Update (HPU) operator, which adaptively refines frequency-specific features and prevents high-frequency detail loss across iterations.

**Strengths:**

The paper introduces a clean and intuitive insight—explicitly separating high- and low-frequency components using Haar wavelets—to address the inherent tension in stereo matching between preserving fine details and ensuring smooth, consistent disparities. The proposed HPU operator is a simple yet impactful mechanism that prevents the blurring of texture details during iterative refinement by decoupling high-frequency context from low-frequency propagation.

**Weaknesses:**

1.Insufficient Evidence on Computational Efficiency: The paper heavily emphasizes faster convergence (e.g., achieving comparable results in 2 vs. 32 iterations) as a key advantage, but it provides an incomplete analysis of the overall computational cost. The introduced wavelet decomposition, dual-branch feature extraction, and complex HPU operator likely incur significant overhead, and without a comprehensive comparison of total FLOPs, parameter count, or inference time against baselines, the claim of superior efficiency remains unsupported and potentially misleading.

2.Limited Novelty in Core Concept: The fundamental idea of processing high and low-frequency information separately is not novel. The paper itself cites contemporary works like Selective-Stereo (Wang et al., 2024) and DLNR (Zhao et al., 2023), which explicitly propose very similar frequency-domain or feature-decoupling strategies to address the same core problem of detail loss in iterative stereo matching. While the wavelet-based implementation and HPU are new, the overarching principle of frequency-specific processing is an incremental step.

3.Unclear Practical Significance and Generality: The performance gains, while statistically leading on benchmarks, are often marginal in absolute terms (e.g., EPE reduction from 0.37 to 0.36 on Scene Flow). The paper does not convincingly argue that these incremental improvements translate to a significant, tangible impact in real-world applications like autonomous driving or robotics, especially given the potential increase in model complexity.

4.Inadequate Ablation on the Wavelet Choice: The entire framework is built upon the Haar wavelet for decomposition, but the paper provides no justification or ablation for this critical design choice. It remains unclear whether the observed benefits stem from the proposed architecture or are specific to the properties of the Haar wavelet, and if other wavelet families or even simple filter banks could achieve similar results.

**Questions:**

1.Given the significant architectural additions (wavelet decomposition, dual-branch feature extraction, HPU), what is the total inference time and FLOPs compared to the baseline RAFT-Stereo and MonSter? The claim of efficiency based on fewer iterations is weakened without this holistic runtime analysis.

2.Why was the Haar wavelet specifically chosen for decomposition, and have you ablated this choice against other wavelet families (e.g., Daubechies) or even simple high-pass filters? The entire method's foundation rests on this choice, yet its justification and sensitivity are not explored.

3.The HPU incorporates both an IFA and an HP-LSTM. Could you provide an ablation study that quantifies the individual contribution of the LSA and HSA modules within the IFA? This would clarify if the performance gain comes from the adaptive fine-tuning or if a simpler fusion mechanism would suffice.

---

### Official Review · Reviewer_EMKG · 2025-10-27

**Soundness:** 3
**Presentation:** 3
**Contribution:** 2
**Rating:** 4
**Confidence:** 4

**Summary:**

In this paper, the authors targets the frequency convergence inconsistency in existing iterative stereo matching methods. The authors observe that high-frequency regions (e.g., edges) and low-frequency regions (e.g., flat regions) converge at different rates during iterative updates, leading to suboptimal performance. To address this issue, the authors develop a plug-and-play module, termed Wavelet-Stereo, to explicitly processes high-frequency and low-frequency components using a dual-branch architecture. Experiments are conducted on diverse benchmark datasets and the impressive results demonstrate the superiority of the proposed method.

**Strengths:**

- Impressive results are achieved on multiple benchmark dataset.
- The proposed method is well motivated.

**Weaknesses:**

- The major concern is about the technical novelty of the proposed method. Incorporating frequency-based ideas into neural networks for image processing has been widely investigated in a wide range of tasks. In the area of stereo matching, a couple of approaches have been developed (e.g., Waveletstereo). Consequently, the difference

- While the proposed method produces impressive performance, its additional cost is not fully discussed. In Table 5, only runtime is presented. More analyese in terms of FLOPs and memory cost should be included such that readers can be clearly aware of the cost for the proposed method and the accuracy-efficiency trade-off.

- The proposed method employs Haar wavelets for frequency-based decompostion without much discussion. It would be better to conduct ablation experiments on different decomposition methods (e.g., other wavelet approaches or Fourier transform).

- In the ablation study, I wonder the performance if the high-frequency branch is removed with only low-frequency branch being employed.

**Questions:**

See Weaknesses

---

### Official Review · Reviewer_jraN · 2025-10-29

**Soundness:** 3
**Presentation:** 2
**Contribution:** 2
**Rating:** 2
**Confidence:** 4

**Summary:**

This paper identifies a key limitation in iterative stereo matching methods, **frequency convergence inconsistency**, where high-frequency and low-frequency regions converge at different rates, leading to performance degradation. To address this, the authors propose Wavelet-Stereo, a plug-and-play module that decomposes the input image into high- and low-frequency components using Haar wavelet transforms. The framework employs a dual-branch architecture: a high-frequency branch using a U-Net to capture fine details, and a low-frequency branch that refines smooth regions iteratively. A novel **High-frequency Preservation Update (HPU)** operator, consisting of an **Iterative Frequency Adapter (IFA)** and a **High-frequency Preservation LSTM (HP-LSTM)**, is introduced to adaptively fuse frequency-specific features while preventing high-frequency degradation across iterations. Extensive experiments show that Wavelet-Stereo achieves state-of-the-art results on multiple benchmarks (KITTI, ETH3D, SceneFlow) and significantly reduces the number of iterations required for convergence.

**Strengths:**

1. The paper introduces the concept of frequency convergence inconsistency, a previously underexplored issue in iterative stereo matching, and provides both theoretical analysis and empirical evidence to support this claim.

2. The proposed Wavelet-Stereo module is lightweight, plug-and-play, and compatible with existing iterative models (e.g., RAFT-Stereo, MonSter). It demonstrates consistent improvements across multiple benchmarks and strong zero-shot generalization ability.

3. The method achieves 1st place on several public leaderboards (KITTI 2012/2015, ETH3D, SceneFlow) and includes thorough ablation studies to validate the contribution of each component (e.g., IFA, HP-LSTM, multi-scale feature extractor).

**Weaknesses:**

1. **Lack of Comparison with Recent Methods**: The paper does not compare with recent foundational stereo models such as FoundationStereo[1] and Stereo Anywhere[2], which limits the reader's understanding of how it performs against the most recent and powerful baselines.

2. **Appendix Figure Layout Issues**: The appendix contains noticeable formatting problems, including missing references to Figure 11 and 12, which disrupts the flow and completeness of the qualitative results section.

3. **Limited Discussion on Computational Overhead**: While the author claims that the proposed Wavelet-Stereo reduces the number of iterations, the additional wavelet decomposition and dual-branch processing introduce extra computational cost, which is not thoroughly analyzed or compared against real-time requirements for applications like autonomous driving.

[Reference] \
[1] BW Wen, M Trepte, et al. CVPR 2025 FoundationStereo: Zero-Shot Stereo Matching. \
[2] L Bartolomei, F Tosi, et al. CVPR 2025 Stereo Anywhere: Robust Zero-Shot Deep Stereo Matching Even Where Either Stereo or Mono Fail.

**Questions:**

1. Given the emergence of recent foundational stereo models like **FoundationStereo**[1] and **Stereo Anywhere**[2], which claim strong generalization and performance, how does your method compare in terms of accuracy, generalization capability, and efficiency on standard benchmarks? Were any experiments conducted with these models for a more comprehensive comparison?
2. While the proposed Wavelet-Stereo reduces the number of iterations, the introduction of wavelet decomposition and a dual-branch architecture likely increases per-iteration computational cost. Can you provide a more detailed analysis of the total inference time, memory footprint, and computational complexity compared to real-time capable stereo methods, especially in the context of autonomous driving applications?
3. Could you explore any optimizations or lightweight variants of your wavelet-based framework to balance performance and computational overhead, particularly for deployment in resource-constrained or real-time scenarios?

---

### Official Review · Reviewer_nQJk · 2025-10-31

**Soundness:** 2
**Presentation:** 2
**Contribution:** 3
**Rating:** 4
**Confidence:** 3

**Summary:**

This paper introduces Wavelet-Stereo, a new plug-and-play framework that addresses the frequency convergence inconsistency problem in iterative stereo matching methods like RAFT-Stereo. By decomposing input images into high- and low-frequency components using Haar wavelets and processing them via a dual-branch architecture, the model enables synchronous optimization of both fine-grained and textureless regions. The proposed method achieves good performance on stereo benchmarks including Scene Flow, KITTI, and ETH3D, while also improving convergence speed and zero-shot generalization.

**Strengths:**

1.	Despite some missing baselines, the experimental protocol is well-structured and follows standard practice, covering major benchmarks such as Scene Flow, KITTI, and ETH3D.
2.	The paper provides a sufficiently detailed analysis of how different frequency components (high vs. low) behave during iterative optimization, offering an interesting perspective for understanding and improving convergence in stereo matching networks.

**Weaknesses:**

1.	**Missing Baselines**: The paper would benefit from including comparisons with recent strong baselines such as FoundationStereo (CVPR 2025) and S²M² (ICCV 2025). These models are highly relevant and would help position the proposed method more convincingly within the current state of the art.
2.	**Middlebury Benchmark**: Including results on the Middlebury dataset would strengthen the evaluation, as it provides valuable insights into model performance on high-resolution indoor scenes.
3.	**Motivation and Evidence**: The claim of frequency convergence inconsistency could be supported with more qualitative or quantitative evidence. While Figure 1 provides an example, additional visualizations or benchmark-level analyses would make the motivation more convincing. The explanation of receptive field behavior currently feels somewhat speculative.

**Questions:**

1. **Model Design**: Figure 4 presents a standard multiscale U-Net-like structure. It may be helpful to clarify the architectural novelty or consider simplifying the figure if it does not highlight a key contribution.

**Details Of Ethics Concerns:**

None.

---

### Note · Authors · 2025-11-12

I have read and agree with the venue's withdrawal policy on behalf of myself and my co-authors.